# Circulating Interleukins as Biomarkers in Non-Small Cell Lung Cancer Patients: A Pilot Study Compared to Normal Individuals

**DOI:** 10.3390/diseases12090221

**Published:** 2024-09-18

**Authors:** Wei-Wen Lim, Jason H. Leung, Chen Xie, Angelina W. T. Cheng, Liping Su, Luh-Nah Lum, Aishah Toh, Siew-Ching Kong, Angela M. Takano, Derek J. Hausenloy, Yang C. Chua

**Affiliations:** 1National Heart Research Institute of Singapore, National Heart Center Singapore, Singapore 169609, Singapore; xie.chen@nhcs.com.sg (C.X.); angelina.cheng.w.t@nhcs.com.sg (A.W.T.C.); su.liping@nhcs.com.sg (L.S.); derek.hausenloy@nhcs.com.sg (D.J.H.); 2Program in Cardiovascular and Metabolic Disorders, Duke-National University of Singapore, Singapore 169857, Singapore; 3Department of Cardiothoracic Surgery, National Heart Center Singapore, Singapore 169609, Singapore; jason.leung.hongting@singhealth.com.sg; 4Clinical and Translational Research Office, National Heart Center Singapore, Singapore 169609, Singapore; lum.luh.nah@singhealth.com.sg (L.-N.L.); aishah.toh@nhcs.com.sg (A.T.); kong.siew.ching@nhcs.com.sg (S.-C.K.); 5Department of Anatomical Pathology, Singapore General Hospital, Singapore 169608, Singapore; angela.takano@singhealth.com.sg; 6Yong Loo Lin School of Medicine, National University Singapore, Singapore 117597, Singapore; 7The Hatter Cardiovascular Institute, University College London, London WC1E 6HX, UK

**Keywords:** interleukin-11, biomarkers, cytokines, non-small cell lung cancer, plasma, serum, enzyme-linked immunosorbent assay

## Abstract

Identifying biomarkers in non-small cell lung cancer (NSCLC) can improve diagnosis and patient stratification. We evaluated plasmas and sera for interleukins (IL)-11, IL-6, IL-8, IL-17A, and IL-33 as biomarkers in primary NSCLC patients undergoing surgical treatment against normal volunteers. Exhaled-breath condensates (EBCs), a potential source without invasive procedures, were explored in normal individuals. Due to separate recruitment criteria and intrinsic cohort differences, the NSCLC and control cohorts were not well matched for age (median age: 65 vs. 40 years; *p* < 0.0001) and smoking status (*p* = 0.0058). Interleukins were first assessed through conventional ELISA. IL-11 was elevated in NSCLC plasma compared to controls (49.71 ± 16.90 vs. 27.67 ± 14.06 pg/mL, respectively, *p* < 0.0001) but undetectable in sera and EBCs by conventional ELISA. Therefore, high-sensitivity PCR-based IL-11 ELISA was repeated, albeit with concentration discrepancies. IL11 gene and protein upregulation by RT-qPCR and immunohistochemistry, respectively, were validated in NSCLC tumors. The lack of detection sensitivity across IL-6, IL-8, IL-17A, and IL-33 suggests the need for further, precise assays. Surprisingly, biomarker concentrations can be dissimilar across paired plasmas and sera. Our results identified a need to optimize detection limits for biomarker detection and caution against over-reliance on just one form of blood sample for biomarker assessment.

## 1. Introduction

Lung cancer is a leading cause of cancer-associated deaths worldwide, accounting for 1.8 million deaths each year [1]. Non-small cell lung cancer (NSCLC) comprises 84% of all lung cancer diagnoses, of which the most common subtypes are adenocarcinoma and squamous cell carcinoma, with a poor 5-year survival of 26% [2]. NSCLC is a highly aggressive disease with a median survival of <10 to 30 months in patients with advanced-stage NSCLC without or with targeted therapy, respectively [3], but also a poor 5-year cancer-specific survival rate of 21% at early stages if untreated [4]. Therefore, early detection and diagnosis is critical to maximize NSCLC patient survival by shifting the disease population to earlier stages, which corresponds with a better prognosis [5,6].

The recent emergence of immunotherapy has altered how advanced NSCLC is being treated. Combinatory immunochemotherapy regiments have been shown to expand the treatable NSCLC population and prolong overall survival, even in patients with advanced disease responsive to immunotherapy [7]. Immunohistochemistry confirmation for programmed death-ligand 1 (PD-L1) expression is the current standard for identifying NSCLC patients more likely to respond to immunotherapy; however, many patients with high PD-L1 expression remain non-responders or suffer disease deterioration with potential multi-organ immunotoxicities [8,9,10,11]. Given this, predictive peripheral blood-based and/or tissue-based biomarkers that can facilitate patient stratification for potential responders against the non-responders prior to the start of immunotherapy are crucial to increase therapeutic benefits.

Blood-based biomarkers for prediction and surveillance of immunotherapy response in NSCLC are an active field of research due to ease of sampling as compared to tissue biopsies, and several potential biomarkers, including different forms of PD-L1, non-coding RNAs, immune cells, peripheral cytokines, circulating free DNA, and tumor mutational burden, have recently emerged (as reviewed in [12]). Among these, peripheral cytokines are perhaps the easiest to develop for diagnostic and surveillance purposes, with most assessments conducted by enzyme-based immunosorbent assays (ELISAs) that can be developed and scaled for high-throughput studies. Additionally, many of these cytokines reflect systemic inflammation which has been suggested to predict response and survival in NSCLC patients treated with immunotherapy [13].

Interleukin (IL-) 11, a member of the IL-6 family, is a cytokine with increasing relevance to a variety of cancers, including colorectal [14,15], breast [16], pancreatic [17], among others, and a potential immunotherapeutic target [18]. IL-11 has also been suggested as a diagnostic marker for NSCLC in bronchoalveolar lavage fluid, serum, and exhaled-breath condensates [19,20], and a potential blood biomarker in NSCLC, although the screening sensitivity remains a limitation (as discussed in [21]). For example, circulating IL-11 levels have reportedly been ranging from non-detectable to 200 pg/mL in NSCLC blood [20,22,23], suggesting an unresolved discrepancy in reported values due to differences in patient characteristics, biological sampling sources, and technical differences (see Appendix A). We therefore sought to validate the potential of IL-11 as a diagnostic biomarker in paired plasma and serum of primary NSCLC patients undergoing surgical treatment against normal volunteers in Singapore. Simultaneously, we investigated the circulating levels of other proinflammatory or immunomodulating cytokines previously suggested as potential biomarkers in NSCLC, albeit normal cohort comparisons in those studies were not always available, which includes IL-6 [24], IL-8 [25], IL-17A [26], and IL-33 [27]. Additionally, we assayed the various cytokines in the exhaled-breath condensates (EBCs) of normal volunteers as an alternative sample source that may reflect secretory cytokine landscape in the lung tissue microenvironment and have potential for future studies. We validated, for the first time, IL-11 gene and protein expression in situ by performing RT-qPCR and immunohistochemistry on paired tumor and adjacent normal lung biopsies in NSCLC patients recruited through the National Heart Center Singapore, demonstrating a tumor-centric upregulation of IL-11 in NSCLC.

## 2. Materials and Methods

### 2.1. Study Patient and Volunteer Cohorts

A total of 60 patients scheduled for elective surgical resection for suspected or confirmed lung cancer were recruited by the Clinical and Translational Research Office of the National Heart Center, Singapore between October 2020 and October 2022. Patients were included if they were above 21 years of age and excluded if they were unwilling or unable to provide consent, biopsy histological diagnoses beyond primary lung cancer, past history of chemotherapy, radiotherapy, immunotherapy, and positive diagnosis for hepatitis B, hepatitis C, or HIV. Clinical staging was determined as per the eighth edition of the American Joint Commission on Cancer (AJCC) TNM staging system for lung cancer [28]. Patients that had surgeries rescheduled without blood and tissue sampling were not assessed for blood cytokines and tissue gene expression, respectively. Not all patients consented to provide blood and/or tissue biopsy; some samples were insufficient for research purposes, or histological diagnosis proved non-NSCLC, resulting in a total sample size of N = 21 for biomarker ELISAs and N = 24 for RT-qPCR studies.

A total of 25 normal volunteers were recruited by the Clinical and Translational Research Office of the National Heart Center, Singapore between April 2022 and April 2023. Inclusion criteria were any able-bodied adult between 21 and 65 years of age and not currently on any long-term medication. Subjects were excluded if they had any instances of previous myocardial infarction, known coronary artery disease, prior cardiac surgery, BMI > 35, alcohol intake >10 units per week, known diabetes mellitus, asthma or chronic obstructive pulmonary disease, current pregnancy, smoker status (including ex- and social smokers), chronic infective disease (including tuberculosis, hepatitis B, hepatitis C, or HIV), prior cancer history, expected life expectancy <1 year, known documented peripheral arterial disease, autoimmune or genetic disease, psychiatric illness, previous stroke, and inability to comply with study protocol.

All patients and volunteers provided informed written consent to participate in the study. The study was approved by the SingHealth Institutional Review Board (2020/2876 and 2022/2149).

### 2.2. Blood Collection

Between 3 and 5 mL of fasted venous blood was collected by a trained phlebotomist at the National Heart Center, Singapore. Volunteers were advised to avoid any strenuous activities prior to their appointment. Whole blood was collected on the same day into (1) BD Vacutainer^®^ SST™ tubes for serum collection or (2) BD Vacutainer^®^ K2EDTA tubes for plasma collection and transferred to the laboratory at room temperature within 15 min. Serum tubes were incubated at room temperature for at least 30 min to induce clotting prior to processing. Plasma tubes were processed within 15 min from collection. Tubes were then centrifuged at 2000× *g* for 10 min at 4 °C. Then, the plasma and serum supernatants were collected separately and transferred into 0.2–0.4 mL aliquots for storage in a −80 °C freezer until analyses were conducted.

### 2.3. Exhaled Breath Condensate (EBC) Collection

EBCs were collected using the disposable collection system (RTube, Respiratory Research, Austin, TX, USA) according to manufacturer’s instructions in the normal volunteer cohort only. EBC collection was conducted consecutively on the same day as their blood collection as described above. Briefly, volunteers had their noses pegged and were instructed by the Clinical Research Coordinator to breathe through their mouth only and maintain their exhalation through the mouthpiece that is connected to the collection tube. After a few minutes of familiarization with the detachable mouthpiece, exhaled breath was collected for 10 min via a one-way valve tube system wrapped by a cooling sleeve for condensation. The collection tube (with the cooling sleeve) was then transferred to the laboratory on ice within 15 min before extraction with a plunger mechanism into 0.2–0.4 mL aliquots for storage in a −80 °C freezer until analyses are conducted.

### 2.4. Enzyme-Linked Immunosorbent Assay (ELISA)

The following assay kits were used according to manufacturer’s instructions: human IL-11 (ELH-IL11, RayBiotech, Norcross, GA, USA; detection range: 3–800 pg/mL), IL-6 (430507, Biolegend; detection range: 7.8–500 pg/mL), IL-8 (ELH-IL8, RayBiotech, Norcross, GA, USA; detection range: 0.8–600 pg/mL), IL-17A (433917, Biolegend, San Diego, CA, USA; detection range: 3.9–250 pg/mL), and IL-33 (435907, Biolegend, San Diego, CA, USA; detection range: 15.6 to 1000 pg/mL). For standard ELISA, samples were run in duplicates at 2-fold dilutions for all plasma, serum, and EBC samples. During the IL-11 analysis, we noted that IL-11 was undetectable in all NSCLC and normal volunteers’ serum and EBCs. Therefore, we procured the high-sensitivity human IL-11 IQELISA kit (IQH-IL11, RayBiotech, Norcross, GA, USA; detection range: 0.49–2000 pg/mL) to better assay IL-11 at lower concentrations. This was conducted as per the manufacturer’s instructions on plasma (5-fold dilution), serum (2-fold dilution), and EBC (2-fold dilution) samples run in duplicates at a volume of 25 µL of sample or standard per well. Samples were run at a minimum of 2-fold dilution, due to potential matrix effects which may interfere with assay. ELISA was conducted by researchers (C.X and A.C.W) blinded to the groups. Freeze-thawing of each sample aliquots was limited to 1 cycle. Upon thawing, samples were temporarily stored at 4 °C for simultaneous or consecutive assays conducted within 5 days. The standard curve was generated on GraphPad Prism (v. 9.4.1, San Diego, CA, USA), and values were extrapolated from the 4-parameter logistics curve-fitting algorithm. Reported cytokine concentrations were multiplied in accordance with the fold dilutions as appropriate. We did not observe assay values above the detection limits for the cytokines tested at the stated dilutions. For values that were below detection limits or coefficient of variance exceeding > 20% for duplicate OD readings, the samples were repeated for confirmation at least once. Repeated values below the detection limit were designated as ‘not detected’ and excluded from further statistical analyses.

### 2.5. RT-qPCR

Flash-frozen NSCLC tumor and far-normal lung biopsies were obtained from patients undergoing lung resection procedures at the National Heart Center Singapore. Tissues were homogenized with silica beads in TRIzol reagent (15596026, Invitrogen, Carlsbad, CA, USA) and column-purified RNA extracted with the PureLink RNA Mini kit (12183025, Invitrogen, Carlsbad, CA, USA) for RT-qPCR analysis for IL11 mRNA on the ViiA7 Real-Time PCR system (Applied Biosystems, Foster City, CA, USA). Relative gene expression values were assessed using the 2^−ΔΔCt^ method normalized to GAPDH levels. Primer sequences were as follows: *IL11* forward 5′-GGACCACAACCTGGATTCCCTG-3′, *IL11* reverse 5′-AGTAGGTCCGCTCGCAGCCTT-3′, *GAPDH* forward 5′-CGACAGTCAGCCGCATCTTCTTT-3, and *GAPDH* reverse 5′-CCAAATCCGTTGACTCCGACCTT-3′.

### 2.6. Immunohistochemistry

NSCLC tumor and matched-adjacent-normal lung biopsies were collected from patients undergoing lung resection procedures at the National Heart Center Singapore. Tissue biopsies were processed at the Department of Pathology at the Singapore General Hospital. Formalin-fixed embedded tissue was sectioned at 5 µm for immunohistochemistry. Slides were blocked with 5% normal horse serum, heat-induced antigen retrieval with Reveal Decloaker (RV1000M, Biocare Medical, Concord, CA, USA), immunostained for IL-11 (PA5-95982, 1:2000, Invitrogen, Carlsbad, CA, USA), goat anti-rabbit IgG-HRP (A0545, 1:500, Sigma-Aldrich, Merck Mille Millipore, Singapore) and Impact DAB Substrate kit, Peroxidase (HRP) (SK-4105, Vector Laboratories, Burlingham, CA, USA) following conventional immunohistochemistry techniques. An isotype control antibody (MA5-16385, 1:2000, Invitrogen, Carlsbad, CA, USA) was applied as a negative antibody control. Slides underwent nuclear counterstain with hematoxylin and mounted for visualization.

### 2.7. Statistics

Data are presented as individual counts (N), values and proportion (%) in tables, and median ± interquartile range (IQR) or mean ± standard deviation (SD) in graphs, as indicated in the figure legends. Statistical analyses were performed on GraphPad Prism (v. 9.4.1). Fisher’s exact test was assessed for proportion differences between NSCLC and normal cohorts. Prior data normality was validated using a Shapiro–Wilk test to determine appropriate parametric or non-parametric statistical testing. A two-tailed unpaired t-test was conducted for parametric comparisons or Mann–Whitney non-parametric test for data not normally distributed accordingly. Receiver-operating characteristic curves were generated on GraphPad Prism, and 95% confidence interval was calculated using the hybrid Wilson/Brown method. A Wilcoxon matched-pairs signed rank test was conducted on IL11 gene expression, comparing paired tumors to adjacent normal tissue in individual patients. Concentration values were presented corrected to 2 decimal values, and statistical significance was established at *p* < 0.05 and presented as exact *p*-values.

## 3. Results

### 3.1. Biomarker Assessments in NSCLC Patients

Patient characteristics are listed in Table 1. Compared to normal controls, our NSCLC patients were significantly older (*p* < 0.0001) and composed of more current and ex-smokers (*p* = 0.0058) but were not statistically different for gender and ethnicity presentation (Table 1). Our NSCLC cohort comprised mostly IA2- to IB-staged lung cancer patients (80.95%), reflecting a predominantly early-stage lung cancer cohort.

We evaluated the circulating levels of IL-11 in the plasma and serum of NSCLC patients as compared to normal volunteers. Surprisingly, we observed substantial differences in the detection ability of the current IL-11 kit depending on their source. In the plasma, IL-11 in the NSCLC patients was significantly greater than that of the normal volunteers (mean ± SD: 49.71 ± 16.90 pg/mL vs. 27.67 ± 14.06, *p* < 0.0001; Figure 1A). Only 1 normal volunteer’s plasma IL-11 (4%) was below the detection limit, in contrast to zero of the NSCLC patients, and was not assigned a value (Figure 1B). In contrast, IL-11 was not detected in the serum of either cohort. We evaluated the diagnostic value of plasma IL-11 by ROC curve analysis to distinguish the NSCLC patients from the normal individuals. Plasma IL-11 has a diagnostic value (area under the curve; AUC) of 0.8267 (95% CI 0.7054 to 0.9480; *p* = 0.0001; Figure 1C).

In comparison, IL-6 levels were detected in serum of NSCLC patients and normal volunteers variably, but undetectable in plasma samples. It has been reported that most IL-6 immunoassays only recognize the unbound IL-6 form and that plasma concentrations are generally undetectable [29], which may explain our non-detection in plasma of either cohort. Serum IL-6 in NSCLC patients was increased, although not statistically significant, as compared to normal volunteers (mean ± SD: 7.94 ± 8.48 pg/mL vs. 2.71 ± 2.36, *p* = 0.0693; Figure 1D). Serum IL-6 levels below the limit of detection were 64% and 42.86% of normal volunteers and NSCLC patients, respectively (Figure 1E).

As elevated serum IL-6, IL-8, and IL-11 levels correlate to worse survival in cancer patients [30], we sought to measure IL-8 in both the serum and plasma samples in the NSCLC patients compared against normal volunteers. We observed that plasma IL-8 was significantly but mildly elevated in the NSCLC plasma (*p* = 0.0413), but mildly decreased in the serum (*p* = 0.0453; Figure 2A). The observation of decreased IL-8 in NSCLC sera could potentially be due to the larger proportion of under-detection in normal compared to NSCLC subjects (56% vs. 23.81%, respectively; Figure 2B), as opposed to accurate differences in assayed values.

IL-17A is an inflammatory cytokine that contributes to Kirsten rat sarcoma viral oncogene (KRAS)-driven lung tumor progression in mice [31]; we therefore sought to compare circulating levels of IL-17A in our cohorts. Despite 100% of plasma samples in either cohort recording a detectable IL-17A concentration, plasma IL-17A was unchanged in our NSCLC cohort compared to controls (Figure 2C). Additionally, no serum sample recorded an IL-17A concentration above the lowest detection limit of 3.9 pg/mL, suggesting plasma samples may be more suitable for assessing IL-17A levels.

IL-33, a secreted alarmin cytokine that plays an important role in type II innate immunity, is known to be upregulated in NSCLC patients associated with tumor malignancy [32]. We observed a trend towards reduction in plasma and serum IL-33 in NSCLC patients compared to controls (Figure 2D). However, this was likely due to the wider variation in recorded IL-33 levels in addition to a larger proportion of NSCLC samples (42.9%, compared to 4–12% in controls, respectively) that was below the limit of detection of the IL-33 ELISA kit (Figure 2E).

### 3.2. EBC Biomarker Assessments in Normal Individuals

Of the 25 normal volunteers, we did not recover sufficient EBCs from one subject due to suspected device leakage, resulting in a total sample size of N = 24. We extracted exhaled-breath condensates (EBCs) in normal volunteers with the aim to establish normal reference ranges of IL-11, IL-6, IL-8, IL-17A, and IL-33 in the EBCs in an unaffected population (Table 2). We note that IL-11 and IL-33 were not detectable in all 24 normal EBCs. In comparison, only one patient (4%) recorded a sufficiently high level of IL-6 and IL-17A to be detected. Lastly, IL-8 levels were detectable in 75% of the normal cohort, with a detectable range between 2.83 and 8.62 pg/mL.

### 3.3. High-Sensitivity IL-11 Assay in NSCLC Patients

Next, we repeated attempts to assay IL-11 at even higher sensitivities, which has been gaining research traction as a diagnostic marker for lung cancer [21], in all three sample types: plasma, serum, and EBC. We observed that sera and EBCs of normal and/or NSCLC cohorts, which were previously undetectable with the conventional IL-11 ELISA, were now detectable in the high-sensitivity IL-11 PCR-based ELISA kit, albeit at ~10-fold difference lower compared to plasma (Figure 3A,B). However, the extrapolated values from the standard curve in this assay were vastly different from the conventional assay (Figure 1), which may have been due to differences in sample dilutions in different assay buffers, PCR-based amplification for higher sensitivity at lower concentrations as compared to chromogen assay, and the smaller sample volumes in the high-sensitivity assay. Correspondence to the vendor’s technical support was unable to resolve the discrepancy, although it was noted that head-to-head comparisons were not performed, and validations of each assay were conducted independently. Notwithstanding, we observed a significant increase in plasma and serum IL-11 in the NSCLC patients compared to the controls (both *p* < 0.05). Plasma IL-11 has a diagnostic value (AUC) of 0.7755 (95% CI 0.6393 to 0.9116; *p* = 0.0012) with similar trends to the conventional assay (Figure 1C), whereas serum IL-11 has a lower diagnostic value of 0.6736 (95% CI 0.5196 to 0.8277; *p* = 0.0417) (Figure 3C). The median IL-11 concentration in normal EBCs was 16.72 pg/mL (IQR 6.1–25.8 pg/mL), which was comparable to previously published values for controls [20]. 

### 3.4. IL-11 Expression in NSCLC Lung Tumor Biopsies

Patient demographics for *IL11* gene expression studies are presented in Figure 4A. Collectively, *IL11* mRNA was >2-fold greater in tumors compared to the adjacent normal biopsies (Figure 4B). Surprisingly, few subjects demonstrated greater *IL11* mRNA in normal, compared to tumor, regions that may suggest some degree of heterogeneity in NSCLC. It is unknown if these may be contributed by other concomitant pulmonary pathologies without a normal lung comparison. As increases in circulating IL-11 levels may be independent of the tumor microenvironment [21], we performed immunohistochemistry for IL-11 protein confirmation in lung tumors and adjacent normal biopsies in two NSCLC patients (Figure 4C). While sporadic expression of IL-11-positive cells were observed in adjacent normal biopsy, IL-11 positivity was more commonly observed in epithelial-like cells in tumor regions. Thus, demonstrating that IL-11 was upregulated specifically within the tumors and may potentially contribute to upregulated circulating IL-11, as seen previously (Figure 1A–C and Figure 3). 

## 4. Discussion

Recent emphasis of appropriate biomarker identification to aid NSCLC diagnosis and surveillance of immunotherapy has spurred new research interests into inflammatory mediators, including interleukins and exosomes which comprise extracellular vesicles containing cell-derived proteins, DNA, messenger and non-coding RNAs such as microRNAs, and long non-coding RNAs [33,34,35]. Exosomal biomarkers present a promising research area for lung cancer diagnosis and therapy. However, its application to the clinic is limited by the need for specific isolation method standardization, heterogeneity in exosome detection methods and characterization, and a lack of reproducibility across studies and understanding of the functional mechanisms by which exosomes target cell-specific cargo delivery [34,35,36,37]. Alternatively, interleukins are well-validated mediators of the immune response and potential therapeutic targets for immunotherapy in cancer treatment [38]. Tumor progression is associated with immune evasion and the presence of chronic unresolved inflammation in the tumor microenvironment where inflammatory cytokines and interleukins play an active role in modulating the immune responses [39] and therefore functions as both a surveyor of the tumor microenvironment and potential immunotherapy target.

In the present study, we examined the circulating levels of IL-11, IL-6, IL-8, IL-17A, and IL-33 cytokines in the blood, differentiated for both plasma and serum, using conventional ELISA techniques. Across the various cytokines, plasma IL-11 demonstrated the most robust assay, with most samples recording a detectable concentration apart from one normal volunteer sample. However, most of the other assays recorded a substantial proportion of readings below the minimal detection range in both NSCLC and normal controls, precluding their current utility as a diagnostic marker. In the last decade, EBCs have emerged as a promising non-invasive, easy-to-collect alternative to more invasive methods, such as bronchoalveolar lavage and/or bronchial biopsies, for examining local airway inflammation [40,41]. We note that EBCs present cytokines at much lower concentrations than blood-based sources (often not detectable for IL-11, IL-6, IL-17A, and IL-33 by conventional ELISAs) and may therefore require more sensitive assays. Utilizing a high-sensitivity IL-11 PCR-based ELISA, we were able to demonstrate detectable IL-11 concentrations across all samples (unlike the conventional ELISA), and significantly higher levels were observed in both the plasma and serum of the NSCLC patients. Lastly, in the NSCLC patients, we observed increased IL-11 gene and protein expression in tumor compared to their adjacent normal tissue biopsies.

Although not being able to detect IL-6, IL-8, and IL-33 in a substantial proportion of samples, this may be reflective of a less-severe, clinically staged NSCLC cohort. Elevated plasma IL-6 and IL-8 are associated with poor prognosis in advanced NSCLC patients undergoing targeted radiotherapy combined with immunotherapy [42]. Likewise, elevated levels of IL-6, but not IL-8, correlate to poorer overall survival in NSCLC [43]. Although we saw an increased serum IL-6 trend in our NSCLC patients, these results must be interpreted with caution due to ~50% of serum and 100% of plasma samples not reporting a detectable reading. Serial changes in serum IL-8 levels correlate with responsiveness to anti-PD-1 blockade therapy in NSCLC patients [25]. Reported median baseline IL-8 levels in non-responders and best responders were 12 and 20 pg/mL, respectively, which were higher than our observations. Additionally, interquartile range (Q1–Q3) for non-responders ranged between 0 and 42 pg/mL, which suggests the inclusion of data below the assay detection limit, whereas we excluded these in our study.

Plasma IL-17A levels in NSCLC patients, at similar concentrations observed in the current study, have been reported to correlate with pneumonitis onset in patients with Stage IIIB-IV NSCLC undergoing immunotherapy [26]; however, a comparison to normal controls were not provided. Alternatively, elevated serum IL-17A has been reported in small cell lung cancer patients at mean levels of 24 pg/mL compared to 12 pg/mL in controls [44] and at comparable levels to other studies with mixed-staged NSCLC cohort [45,46,47]. This contrasts with our paired plasma–serum comparison, where we observed no detectable serum IL-17A in NSCLC and normal control cohorts. It should be noted that in a study with 15 stage-I-IIA NSCLC and 30 normal controls, elevated EBC IL-17A was observed in NSCLC patients compared to normal controls, albeit at lower concentrations (<3 pg/mL) [48], which is lower than the detection limits of the kit used in this study. Even with the more sensitive assay, detectable, positive serum IL-17A was observed in only 20% of samples [48], thereby limiting their assessment of IL-17A in serum.

Serum and EBC IL-11 concentrations have been reported to be increased in NSCLC patients, correlating to the severity of tumor staging [20]. Detectable serum IL-11 in NSCLC (ranging from 123 to 324 pg/mL) was increased as compared to controls (ranging from 23 to 77 pg/mL), whereas reported EBC IL-11 levels were lower as compared to serum but still elevated in NSCLC (ranging from 26 to 76 pg/mL) compared to controls (8 to 17 pg/mL). However, it should be noted that IL-11 levels have also been non-detectable in plasma or serum in patients with lung pathologies, including bronchial carcinoma [22]. In bronchoalveolar lavage fluid (BALF), IL-11 levels were higher in lung adenocarcinoma (median 107 pg/mL) as compared to controls, with negative detection for IL-11 solely to be ~10 to 20% in lung cancer patients depending on the cohorts [19]. We therefore chose, for the current study, the current conventional and high-sensitivity IL-11 ELISAs with minimal detection limits of 3 and 0.49 pg/mL, respectively. In our study, high-sensitivity IL-11 IQELISA reported higher levels in plasma and serum in NSCLC compared to normal volunteers (refer to Figure 3). This potentially suggests that IL-11 may be used as both a diagnostic and monitoring tool for treatment responses in NSCLC, akin to CEA as a traditional tumor biomarker [20]. Recently, astrocyte-derived IL-11 has been demonstrated to upregulate PD-L1 expression and promote immune escape in NSCLC by reducing T lymphocytes demonstrating direct immunomodulating effects of IL-11 and a potential therapeutic target [49]. 

Through this pilot study, we demonstrated that high-sensitivity IQELISA can detect IL-11 in the EBCs of normal individuals, although corresponding NSCLC samples were lacking and of interest for future studies. Notably, except for plasma, IL-11 could not be detected using conventional ELISA methods in EBC nor serums of normal individuals, suggesting the need of precise assays for small differences in the lower ranges. Importantly, we were able to probe for IL-11 gene and protein expression in NSCLC tumors compared to their adjacent normal controls, providing paired comparisons in the same patient. To the best of our knowledge, this is the first-time IL-11 protein immunostaining has been performed in situ and in paired-adjacent normal and tumor regions in NSCLC patients. Our results agree with others [50], suggesting a mechanistic role of IL-11 in lung cancer. The application of these findings will need to be explored in different TNM-staged NSCLC patients to determine correlation with the disease severity. 

IL-11 is known to stimulate proinflammatory responses, including IL-6 and IL-33, through the JAK–STAT pathway in tissue fibroblasts [51], of which cancer-associated fibroblasts contribute centrally to the tumor microenvironment. Serum IL-33 has been proposed as a diagnostic and prognostic marker for NSCLC [27]. Using a cut-off value of 68 pg/mL (95% specificity in normal volunteers), Hu et al. found serum IL-33 to have a diagnostic value of 0.736 AUC for NSCLC and high IL-33 levels at baseline correlated with poorer prognosis. Contrastingly, Kim et al. reported opposing results with initial elevated plasma IL-33 levels only in stage I lung cancer patients compared to normal controls but decreases with lung cancer stage progression [52]. Likewise, we also noted a trend towards reduced IL-33 in our NSCLC cohort (despite the large data variability) and a bigger proportion of NSCLC samples that were under-detected. Notably, the reported IL-33 concentrations by Kim et al. were in line with our current findings in both plasma and serum (in the ng/mL range) rather than that of Hu et al. (in pg/mL). The reason behind this discrepancy is not known but likely attributed to differences in ELISA kit vendors and their accompanied detection ranges.

The reasons behind detectability differences in IL-6, IL-11, and IL-17A between plasma and serum reported here are incompletely understood. Both plasma and serum comprise the liquid component of blood following removal of blood cells, with differences in the clotting process to obtain serum that is prevented in plasma with anticoagulation. Poor agreement between plasma and serum cytokines in paired subjects has been previously reported [53], where inflammatory cytokines are generally higher in serum than plasma [54]. Additionally, IL-11 is known to upregulate the von Willebrand factor, a crucial clotting factor mediating hemostasis in humans [55]. Whether circulating levels of IL-11 may be altered through the coagulation process for serum preparation remains unexplored.

### Study Limitations

Several limitations exist in our study. Firstly, our sample size was small, and the normal volunteer cohort was not age-matched to our NSCLC cohort consisting largely of early-stage NSCLC. This was due to differences in the inclusion/exclusion criteria in the separate recruitment processes; we selected mostly younger normal volunteers that were non-smokers without any known diseases, whereas recruitment for NSCLC patients was based on clinical presentation for surgery. Additionally, EBCs were provided by the normal cohort in an attempt to derive reference-normal ranges on interleukin concentrations but were unavailable in NSCLC patients due to recruitment order. We highlight that conventional ELISA sensitivities commonly performed for plasma or serum are not optimized towards the lower concentrations expected in EBCs, often presented below assay detection limits, and re-enforce the need for higher-sensitivity assays to meaningfully study EBCs in health and disease. Secondly, each analyte was assessed by only one commercially available ELISA kit, and it is possible that other kits, if employed, may present differences, as has been suggested for IL-17A, with different percentages of detectability [56]. Furthermore, IL-11, like IL-6, can form cytokine–receptor complexes extracellularly, which can potentially affect their detectability in immunoassays that only detect the free or unbound cytokine [29]. All the kits used were validated for sensitivity and specificity with recombinant peptides (free form) according to their respective datasheets and do not necessarily account for trans-signaling mechanisms. This is performed by spiking recombinant peptides into samples and assessing for assay recovery, which may not necessarily reflect endogenous peptides’ properties entirely, depending on the expression system employed [57]. Thirdly, we were surprised by the differences between plasma and serum IL-11 concentrations. Despite the abnormally high IL-11 concentrations (in plasma; ng/mL) observed with the high-sensitivity IQELISA, we report trends with highest concentrations observed in plasma, serum following, and least in the EBC, which explains the non-detectability by conventional assays. Furthermore, as reagents in both kits are proprietary information, we are unable to assess if they recognize similar epitopes which may affect detection. However, the reported plasma IL-11 concentrations in our controls were comparable to others utilizing conventional assays [17,58,59]. Conversely, our reported IL-11 concentrations by the IQELISA kit were comparable to levels previously reported in serum and EBCs from NSCLC and normal individuals using a conventional kit from a different company [20], but plasma samples were not explored in that study. Whether this discrepancy may also be due to the choice of blood collection tubes, as is recently seen with specific proteins and metabolites, is currently unknown [60]. Recent developments in ultra-sensitive IL-11 assays have allowed for even lower limits of quantification (0.006 pg/mL) and will be especially useful in assessment of antibody target engagement biomarkers for clinic trials [61]; however, these assays are not commercially available for widespread use. We also note that at the time of drafting the manuscript, the IL-11 IQELISA kit has not been cited before and would encourage more replication in the scientific community for reproducibility.

## 5. Conclusions

Changes in biomarker levels may potentially aid prediction and surveillance of immunotherapy responses in NSCLC. In our small cohort study, pilot data suggest that IL-11 remains a potential and robust blood-based biomarker that is elevated in NSCLC patients, more so than IL-6, IL-8, IL-17A, and IL-33, which have variable detectability across sample sources. Nevertheless, further development and refinement are warranted in the field to ensure robust specificity and sensitivity across assay modes, to the individual assay’s detection limits, biosample sources, and reproducibility of IL-11 measurements across platforms. Lastly, the use of blood-based and/or circulating biomarkers as a surveyor of immunotherapy response in NSCLC remains to be optimized and poses an attractive research avenue that warrants future study.

## Figures and Tables

**Figure 1 diseases-12-00221-f001:**
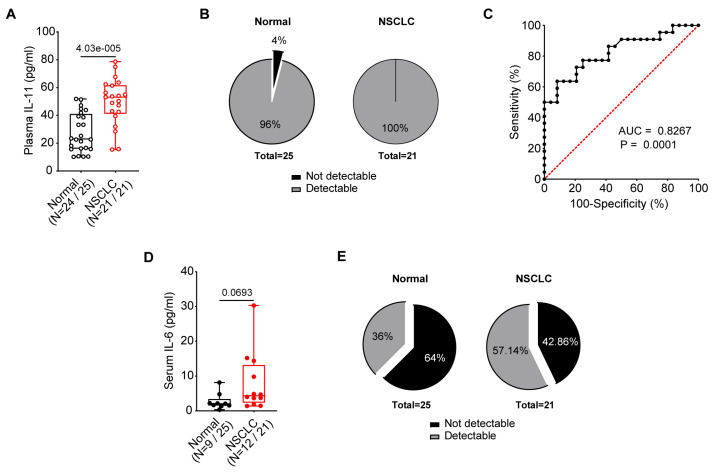
Evaluation of IL-6 family cytokines, IL-11 and IL-6 concentrations, in the blood of normal and NSCLC patients. (**A**) Elevated plasma IL-11 concentrations were observed in NSCLC (N = 21) compared to normal individuals (N = 25). (**B**) Pie-chart depicting the percentage proportion of plasma samples with detectable IL-11 concentrations in NSCLC and normal individuals. One normal individual plasma IL-11 concentration was below the limit of detection. No serum from either cohort reported detectable IL-11 concentrations > 3 pg/mL. (**C**) The ROC curve analysis for plasma IL-11 to distinguish NSCLC patients from normal individuals. The dashed red line represents the random classifier. (**D**) A trend towards increased serum IL-6 was observed in NSCLC (N = 12) compared to normal individuals (N = 9). (**E**) Pie-chart depicting the percentage proportion of serum, with detectable IL-6 concentrations in NSCLC and normal individuals. Sixteen normal and ten NSCLC sera did not report a detectable IL-6 concentration, whereas plasma IL-6 was not detectable across both cohorts. (**A**,**D**) Data presented as median ± IQR with whiskers indicating minimum and maximum values. Two-tailed Mann–Whitney test was conducted for statistical analyses. Open and closed symbols indicate plasma and serum, with normal in black and NSCLC in red, respectively. (**C**) Wilson/Brown method was used to compute the confidence interval for statistical analyses.

**Figure 2 diseases-12-00221-f002:**
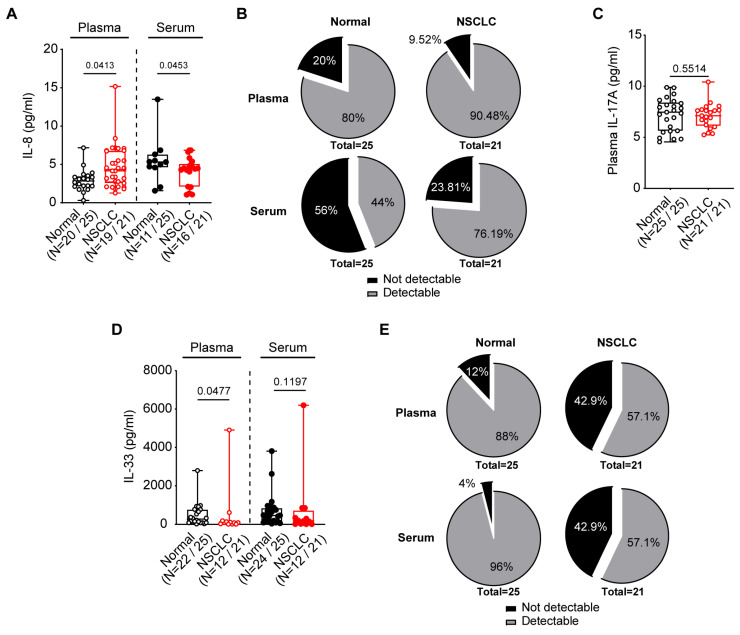
Circulating proinflammatory cytokines, IL-8, IL-17A and IL-33, in the blood of normal and NSCLC patients. (**A**) Elevated plasma, but not serum IL-8, were observed in NSCLC compared to normal individuals. (**B**) Pie-chart depicting the percentage proportion of plasma and serum with detectable IL-8 concentrations in NSCLC and normal individuals. In total, five normal and two NSCLC plasma as compared to fourteen normal and five NSCLC sera IL-8 concentration were below the limit of detection <0.8 pg/mL. (**C**) Plasma IL-17A levels were unchanged between NSCLC and normal individuals, with all plasma samples detected. No serum in either cohort recorded IL-17A concentrations was above the lowest limit of detection of 3.9 pg/mL. (**D**) Plasma and serum trended towards reduced IL-33 levels in NSCLC as compared to normal individuals. (**E**) Pie-chart depicting the percentage proportion of plasma and serum with detectable IL-33 concentrations in NSCLC and normal individuals. Three normal and nine NSCLC plasma as compared to one normal and nine NSCLC sera IL-33 concentration were below the limit of detection <15.6 pg/mL. (**A**,**C**,**D**) Data presented as median ± IQR, with whiskers indicating minimum and maximum values. Two-tailed Mann–Whitney test was conducted for statistical analyses. Open and closed symbols indicate plasma and serum, with normal in black and NSCLC in red, respectively.

**Figure 3 diseases-12-00221-f003:**
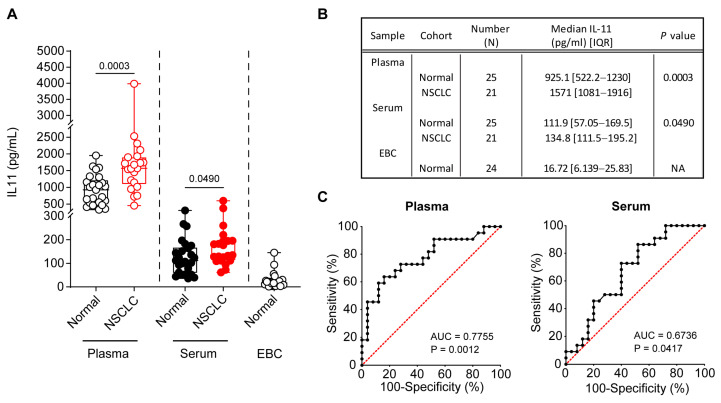
High sensitivity IL-11 IQELISA in plasma, serum and EBC in normal individuals versus NSCLC patients. (**A**) Elevated plasma and serum IL-11 levels were observed in NSCLC compared to normal individuals. In addition to 24 normal EBC, all 25 normal and 21 NSCLC plasma and serum recorded a detectable IL-11 concentration with the high-sensitivity human IL-11 IQELISA kit (IQH-IL11, RayBio). Two-tailed Mann–Whitney test was conducted for statistical analyses. Open and closed symbols indicate plasma and serum, with normal in black and NSCLC in red, respectively. (**B**) Table depicting median IL-11 concentrations and interquartile ranges, respectively. (**C**) The ROC curve analysis for plasma and serum IL-11 to distinguish NSCLC patients from normal individuals. The dashed red lines represent the random classifier. (**A**,**B**) Data presented as median ± IQR with whiskers indicating minimum and maximum values. Two-tailed Mann–Whitney test was conducted for statistical analyses. (**C**) Wilson/Brown method was used to compute the confidence interval for statistical analyses.

**Figure 4 diseases-12-00221-f004:**
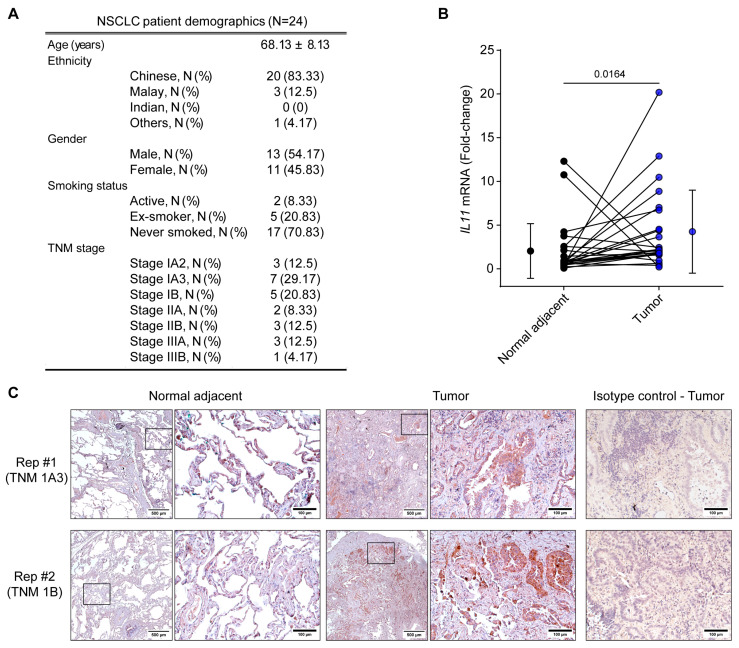
Interleukin-11 is upregulated in NSCLC tumors. (**A**) NSCLC patient demographics (N = 24) used for RT-qPCR for *IL11* gene expression studies. (**B**) *IL11* gene expression was assessed in adjacent normal and tumor biopsies in paired subjects by RT-qPCR (N = 24). Gene expression was normalized to GAPDH and expressed as fold change. Data are presented for paired individual subjects with the collated mean ± SD for each group. Two-tailed Wilcoxon test was conducted for statistical analyses. (**C**) IL-11 expression by immunohistochemistry in NSCLC tumors compared to adjacent normal biopsies in paired subjects (N = 2 biological replicates). Negative control sections were stained with an isotype control antibody.

**Table 1 diseases-12-00221-t001:** Normal volunteers and NSCLC patients’ characteristics for biomarker assessments.

	Normal Volunteers (N = 25)	NSCLC Patients (N = 21)	*p*-Value
**Characteristics**			
Median Age (IQR)	40 (35.5–48.5)	65 (60–70)	5.88 × 10^11^
Gender, Female (%)	12 (48)	7 (33)	0.3769
**Ethnicity**			0.1107
Chinese, N (%)	24 (96)	16 (76.19)	
Malay, N (%)	1 (4)	3 (14.29)	
Indian, N (%)	0 (0)	0 (0)	
Others, N (%)	0 (0)	2 (9.52)	
**Smoking status**			0.0058
Non-smoker, N (%)	25 (100)	15 (71.43)	
Ex-smoker, N (%)	0 (0)	4 (19.05)	
Current smoker, N (%)	0 (0)	2 (9.52)	
**Lung cancer stage**	NA		NA
Stage 0, N (%)		1 (4.76)	
Stage IA1, N (%)		0 (0)	
Stage IA2, N (%)		6 (28.57)	
Stage IA3, N (%)		6 (28.57)	
Stage IB, N (%)		5 (23.81)	
Stage IIA, N (%)		0 (0)	
Stage IIB, N (%)		1 (4.76)	
Stage IIIA, N (%)		1 (4.76)	
Stage IIIB, N (%)		1 (4.76)	

Age was analyzed using Mann–Whitney test for non-normal distributed data. Gender, ethnicity, and smoking status were analyzed with Fisher’s exact test for difference in proportions. Lung cancer status was assessed in accordance with AJCC TNM staging (8th ed.). Percentages were rounded to the nearest two decimal places. IQR, interquartile range; NA, not applicable.

**Table 2 diseases-12-00221-t002:** EBC cytokines by conventional ELISA in normal volunteers (N = 24).

Cytokine (Concentration)	Min	Median	Max	Subjects with Detectable Analytes, N (%)
IL-11 (pg/mL)	ND	ND	ND	0 (0)
IL-6 (pg/mL)	NA	7.64	NA	1 (4.17)
IL-8 (pg/mL)	2.83	3.06	8.62	18 (75)
IL-17A (pg/mL)	NA	27.45	NA	1 (4.17)
IL-33 (pg/mL)	ND	ND	ND	0 (0)

ND denotes not detectable above the lowest limit of detection. NA denotes not applicable, as only one sample recorded a detectable concentration within the detection range.

## Data Availability

The data presented in this study are available on request from the corresponding author.

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
