# Peer review of "Circulating Interleukins as Biomarkers in Non-Small Cell Lung Cancer Patients: A Pilot Study Compared to Normal Individuals"

_diseases, 2024, doi:10.3390/diseases12090221_

Round 1

Reviewer 1 Report (Previous Reviewer 1)

Comments and Suggestions for Authors

First, I would like to complement the authors who with a few simple adjustments redirected this paper to address the negative results they found. The original manuscript presented the non-detection of the various bio-markers in different matrix as an aside which created negative reliability in the presented data. In this manuscript it is presented as the part of the focus of the work, it makes the data far more believable presented in this way. Several issue’s still remain for me though

1.       The introduction covers the theoretical reasoning behind the study adequately, the added section concerning circulating il-11 redirects the focus of the paper nicely. However, this opens the authors up to further criticism, firstly you have drawn a conclusion here that biological sampling sources could be linked to the discrepancies in measurements. You offer no evidence for this, a simple tabulated summery of the references indicating matrix, kit used/manufacturer, concentration limits of the assay, patient characteristics etc would solve this issue.

2.       Secondly, if you know that different matrixes give concentration levels below the detection limits of commercial kits why are you using the same kits? Since the authors should know they were going to get lower levels shouldn’t part of this study be gaining the detection capabilities for the various biomarkers. The authors redirecting to investigating the reliability of the biomarker measurement leads to the question of why did you not address the issue of sensitivity first? It’s a big hole in your arguments. I realise at the time of this study this was not your aim and you are making an effort to make your data publishable, however, the authors should justify the use of kits that are not going to work according to the reasoning derived in the rational of the study.

I direct the authors attention to the following reference, I found this easily and theoretically it solves all of your issues for Il-11, if I can find this why couldn’t the authors? The authors do address this issue using a more sensitive assay but this is not made clear until you read that section, the author also cast doubts about their own data by trying to extrapolate beyond the assay range. See below.

Myzithras, M., Lin, S., Radden, L., Hess Kenny, C., Cai, Z., MacDonald, A., … Hansel, S. (2022). Development of novel ultra-sensitive IL-11 target engagement assays to support mechanistic PK/PD modeling for an anti-IL-11 antibody therapeutic. mAbs14(1). https://doi.org/10.1080/19420862.2022.2104153

3.       Carrying this theme on, in the results section 3.2 the authors state they were surprised in the substantial difference between matrixes, this contradicts the statements in the redirected introduction. It also brings into question again why didn’t the authors investigate more sensitive assays from the start. I direct the authors attention again to the following which measured detectable levels in all plasma sample of normal patients.

Thompson DK, Huffman KM, Kraus WE, Kraus VB. Critical appraisal of four IL-6 immunoassays. PLoS One. 2012;7(2):e30659. doi: 10.1371/journal.pone.0030659. Epub 2012 Feb 9. PMID: 22347395; PMCID: PMC3276568.

4.       Section 3.3 concerns the high sensitivity PCR based assay, this type of assay is not extrapolatable, there is an upper limit of detection as well as a lower limit of detection. These limits are usually quite narrow. I suggest the authors remove this conclusion, it makes their analytical work appear poor, especially linking it to differences in dilutions between different assays. This suggests significant matrix effects which implies poor analytical technique not to know this, any kit will always give the limits to which samples can be diluted to maintain integrity. If the authors stepped outside of those recommendations, then the results are invalid. Noting that the assays used are not cross validated by the vendor is an important point and should be kept.

5.       Section 4 – While the authors discus the results found in an adequate fashion it is not made clear that these compounds are all part of an interconnected network, changing one changes the others. This is really important during ELISA or PCR assays which operate on the free compound’s reactivity, this can negatively effect the accuracy. It has been reported that a LCMSMS assay can measure nearly all cytokines simultaneously without affecting their balance in the network, again this can be a way out of the authors predicament. Sensitivity via LCMSMS is of a broader range and samples can be concentrated more easily increasing sensitivity.

6.       Soluble receptors, cytokines linked to receptors are not detectable by immune-based assays. Up or down regulation of receptors will affect the free concentration of each cytokine which in turn will affect free concentrations. The author cover this briefly but more focus should be made with the relationship between cytokines if possible when discussing the results.

In all, the revisions of this paper have been successful in presenting the data a more positive light, however, the authors should critically look at each section to confirm continuity of the direction of their thoughts.

Comments on the Quality of English Language

Minor grammatical errors here and there but nothing significant.

Author Response

Reviewer 2 Report (Previous Reviewer 2)

Comments and Suggestions for Authors

1. I checked the resubmitted manuscript, and I found the authors have completely revised it.

2. I have no other suggestions.

3.I hope this manuscript can be accepted.

Author Response

We are thankful to the reviewer for his/her suggestions that have helped improved the manuscript overall for publication.

This manuscript is a resubmission of an earlier submission. The following is a list of the peer review reports and author responses from that submission.

Round 1

Reviewer 1 Report

Comments and Suggestions for Authors

An interesting concept with a well prepared manuscript, clear figures and excellent quality of English. However, the methodology is questionable due to nature of the results presented, the data the authors reveled is just plain weird for certain components. This creates an increasing element of doubt in the validity of the work as you read it which is not good. I have many questions that cannot be answered when reading this manuscript some of which I have indicated below.

Point 1 Exhaled breath condensate

What validation criteria did the authors use to verify accuracy/precision and robustness of this sampling technique. They report variable results seen between different matrix or sampling techniques but do not show how they proved reliable sample collection of the EBC. This also is point for the plasma and serum collection, did they verify prior to the start of patient collection that repeat samples for the same patient across different days show similar results. Controlling variables such diet or exercise in the healthy volunteers would be of particular importance since it has been demonstrated that a diet of high fatty meal will decrease IL-6 concentrations significantly while exercise has been shown to increase IL-6 significantly, how would this effect the downstream IL-11 is not actually known.

Point 2

While it is not uncommon to compare a healthy population to a diseased population the fact differences in age and smoking status was significantly different do seem to suggest a non-compatibility in comparison of the data, especially considering the known effects of diet and exercise on one of the related molecules of IL-11. Also the text appears to indicate that the NSCLC patients consisted of more active individuals compared to the health controls. Not sure if that is what the authors meant to say considering the “healthy controls” were younger and non-smokers (no indication given about passive exposure to smoking, could be an important consideration)

Point 3

Differences in IL-11/6 between matrixes such as serum and plasma was very strange, it strongly suggests that there are unknown issues with the analytical technique. More than one research paper has shown that IL-11 can be detected in serum of disease patients groups, so for IL-11 not to be detected in both cohorts creates extreme doubts about the analytical data of all measurements. In a completely inexplicable turn of fate the IL-6 was not detected in plasma but was seen in serum. The authors need to address this issue to infer reliability on the rest of their data because as it stands the authors are comparing plasma IL-11 with serum IL-6, not ideal, whereas a simple literature search revels group who have measured IL-11 and IL-6 in both matrixes. With this in mind how can the Il-8 data be trusted. I have significant problems with this issues, I recognize that sometime analytical issues can create problems but this is just plain strange and cannot in all conscious accept the data as a “real” reflection of the situation. That’s a shame because I’m sure somebody has put a lot of effort into all of this, The PCR based method shown IL-11 serum levels 10 fold lower than the elisa plasma levels, however, the comparison of the plasma data between assay was significantly different. I do not see how to trust the data being presented

Point 4

“The authors appear to be validating their assay by using them in determining study samples, this implies a blind trust in the kit manufacturer, not always a good idea. The authors need to include some positive controls in each assay to confirm viability of the kit used and provide some indication of reliability of the assay data between different data sets.” At least that is what I noted during the review of the paper. It was not until the final few paragraphs of the discussion I came across the quality control data, the authors should note that my initial impression is the one that dictates whether you believe the results or not and consider moving the control data to earlier.

Point 5

The authors quite correctly indicate that different manufacturers of kits give different sensitivities and/or results. However, considering the serious issues of sensitivity that they have experienced between matrixes it is my firm belief they should have done much more investigation into this issue before attempting to publish. A better understanding of the problems would create a firmer degree of trust in the data they are trying to present. At the moment the doubts about the assay leads me to have doubts about their conclusions and whether these are a viable biomarker for this disease. If you cannot trust the results of one manufacturers kit to match another how can a coherent biomarker analysis protocol be established.

Comments on the Quality of English Language

Minor editiorial changes to the english needed in a few places to clear up misunderstandings

Reviewer 2 Report

Comments and Suggestions for Authors

1.This research focused on Circulating interleukins as biomarkers in non-small cell lung cancer patients:

 a pilot study compared to healthy individuals, after check in pubmed,have some references about this topic such as PMID: 30953795、34142593 、27922075, so this manuscript was with some importance and Novelty.

2.Although this manuscript was related with the clinical problem , but still many places should be revised.

3.The whole stucture of this manuscript can be more perfect, such as introduction section was so simple and brief.should foucus on il-11 importance in nsclc and now the limits of this reaeach.

4.The references can be renewable and suggest that many should from the last 3 years maybe much more better.

5. The samples make me confused such as a total sample size of N=21 for biomarker ELISAs and N=24 for RT-qPCR studies.and 25 normal healthy volunteers,but table 1 Healthy volunteers

(N=25) NSCLC patients (N=21) ,figure 1 Elevated plasma IL-11 concentrations were observed in NSCLC

(N=21) compared to normal individuals (N=24).

6.if the il-11 relaed with the prognosis of nsclc,what is your plan in next reseach/

7. The figures should revise such as figure 4 was framed but other figures not.